# Frequent Visits to an Outdoor Range and Lower Areas of an Aviary System Is Related to Curiosity in Commercial Free-Range Laying Hens

**DOI:** 10.3390/ani10091706

**Published:** 2020-09-21

**Authors:** Manisha Kolakshyapati, Peta Simone Taylor, Adam Hamlin, Terence Zimazile Sibanda, Jessica de Souza Vilela, Isabelle Ruhnke

**Affiliations:** 1School of Environmental and Rural Science, Faculty of Science, Agriculture, Business and Law, University of New England, Armidale, NSW 2351, Australia; peta.taylor@une.edu.au (P.S.T.); tsiband2@une.edu.au (T.Z.S.); jdesouz2@myune.edu.au (J.d.S.V.); iruhnke@une.edu.au (I.R.); 2School of Science and Technology, Faculty of Science, Agriculture, Business and Law, University of New England, Armidale, NSW 2351, Australia; ahamlin@une.edu.au

**Keywords:** aviary, behavior, egg, layer, non-cage, personality, poultry, ranging, welfare

## Abstract

**Simple Summary:**

Individual hens’ preferences to spend time at particular locations within free-range aviary housing system may be influenced by personality and behavioral characteristics such as curiosity or fear. We monitored hens’ location for their production life on a commercial free-range farm: time spent in the outdoor range, upper feeder, lower feeder, and nest box tier. At the end of lay, we conducted a series of validated behavioral tests to assess fearfulness and exploration. We hypothesized that if temperament encouraged preferences for particular areas, we would find relationships with early life behavior. Conversely, relationships with choice of space and whole production life ranging would indicate environmental influences on temperament. Whilst we could not determine causation, more time spent on the range and lower feeder tier was associated with increased curiosity but only when considering whole-life not early-life ranging. We found little evidence that fearfulness or individual coping styles was associated with time spent at a specific housing location in early or whole life. Body weight prior to range access was the strongest predictor of more time spent at the lower and less on upper feeder tier. We provide evidence that preference for more complex environments such as range may increase hen curiosity.

**Abstract:**

Individual hen preferences to spend time at particular locations within a free-range aviary system and relationships with temperament is relatively unknown. Hens (*n* = 769) from three commercial flocks were monitored with Radio Frequency Identification technology to determine time spent on the range, upper and lower aviary tiers, and nest boxes. Prior depopulation, novel arena (NA) and novel object (NO) tests assessed exploration and fearfulness. During early life; more time on the lower tier was associated with more lines crossed in the NA test (*p* < 0.05). No other evidence suggested preference during early life was related to fear or curiosity. More time on the range and lower tier were associated with heavier pre-ranging body weight and gain (*p* = 0.0001). Over the hens’ whole life; time spent on range and lower tier was associated with approaching the NO (*p* < 0.01). More time spent on the upper tier was associated with less time near the NO and fewer lines crossed in NA (*p* < 0.01). The relationships during early and whole life use of space and some potential indicators of fearfulness were inconsistent and therefore, no strong, valid, and reliable indicators of hen fearfulness such as freezing were identified.

## 1. Introduction

In free-range and aviary housing systems, individual hen behavior can vary significantly, such that, some hens rarely access the outdoors while others utilize the range regularly [1,2,3,4]. Free-range and aviary housing systems provide hens with horizontal and vertical space, allowing for various choices, opportunities, and interactions; including dust bathing, sun bathing, perching, and exploration which are believed to improve hen welfare [5,6]. However, not all hens access the outdoor range when provided with the opportunity [1,2,4,5,6]. Individual differences in ranging behavior, and preferences for specific areas of the three-dimensional aviary system, may be associated with individual differences in temperament, such as fearfulness [7,8,9,10].

Fear is considered a negative welfare state and has been associated with various factors including social stress, severe feather pecking, social isolation, novel resources, and environments and infrequent or negative human interactions [2,11]. Furthermore, poultry may be fearful of open spaces such as the range [2,6,12,13,14,15]. Research has shown that individual differences in fearfulness varies due to brain morphology and serotonin turnover and is regulated by a combination of genes and environment [16,17]. Experimental and commercial housing systems consisting hens of 37–38 and 41 weeks of age [2,18], behavioral assessments have shown that hens that prefer to stay inside the shed are generally more fearful than the hens that use the range frequently, as evident by more freezing behavior and less movement in open field tests and longer tonic immobility durations [2,18]. Neophobia associated with novel stimuli in the range environment may be an important factor associated with where hens chose to spend their time in free-range housing systems. Fearfulness and distress can impact health and immune function via activation of the neuroendocrine system, the hypothalamic–pituitary–adrenal axis and the autonomous nervous system [19]. Although stress responses can help the animal to cope with the environment and survive in the short term, chronic stress can have detrimental effects resulting in impaired biological functions such as reduced reproduction and growth [19]. As such, long term distress experienced by laying hens is likely to result in reduced body weight and egg production [20,21]. Indeed, minimal range use has been associated with lower body weight, less body weight gain, and poor egg production which may reflect chronic stress associated with hens that stay indoors [3,22,23]. However, such relationships are largely unknown, particularly in commercial conditions over the production life of the hen.

Conversely, novelty (novel environments and resources) has also been shown to motivate hens to explore [24,25]. For example, broiler chickens showed greater motivation to seek novelty by showing a preference for enriched locations containing straw bales, platforms, ramps, compared to empty areas with no novel resources [26]. Curiosity therefore may be a major driver of range use. As positive welfare states are characterized by the engagement with objects that are intrinsically rewarding [27,28], hens that are curious of novel environments, such as an outdoor range, are likely to experience pleasure during exploration [28].

Personality (or temperaments) are reflective of behavioral characteristics, such as exploration and/or boldness which can be expressed by the individual consistently over time and across contexts [7,9,10]. Hens initially fearful of novel environments, such as outdoor range, will explore the range more frequently after repeated exposure and increased familiarity [29,30], suggesting habituation to the range during early life may be an important factor of range use [31]. Furthermore, enriched environments and exercise have been found to be associated with hippocampal neurogenesis and these newly generated cells likely have function in cognition [32]. As such, the use of specific areas in a complex housing system such as an outdoor range may impact temperament in addition to initially encouraging hens to access the range. Curious birds show interest in the objects available in the environment spending more time in exploration [33]. Although, due to the significant negative correlation between fearfulness and curiosity, it is often difficult to tease apart if the impact of environmental complexity is reflective of increased curiosity, a reduction in fear or both [34]. A study by Kozak et al. (2019) [34] showed that the hens with high levels of curiosity/exploration quickly started approaching the novel objects and spent substantial time on exploration in novel arena test equipped with novel objects. It may be that curiosity and fearfulness determine use of the outdoor range, and other areas of an aviary housing system.

Further understanding of the relationships between fear, curiosity, and the use of distinct areas within a commercial hen shed as well as the outdoor range area may help to improve facility design and management techniques to improve the welfare of poultry. The purpose of this study was to investigate the relationship between fearfulness, curiosity, and body weight with use of the range and aviary (lower tier, upper tier, and nest box) areas in commercial free-range hens during their entire production period (18–74 weeks of age). We hypothesized that fearful hens would prefer to stay inside the shed, predominately on elevated areas and would have lower body weight throughout their production cycle indicative of chronic stress. We propose that preferences for location during the first few weeks of range access and fear and curiosity temperament, would provide some evidence that characteristics are drivers of range use. However, preferences over the whole production cycle would provide insight to changes in temperament from use of specific areas.

## 2. Materials and Methods

This research was approved by the University of New England Animal Ethics Committee (AEC17-125).

### 2.1. Experimental Design and Study Population

This experiment was a part of larger experiment using 5 flocks of 40,000 Lohmann Brown hens each where 3125 hens/flock were monitored for their aviary and range use during their life in the hen house from 16 to 74 weeks of age [35]. Thus, hens experienced access to the range during all seasons. The indoor stocking density was 9 hens/m^2^ and outdoor stocking density was 1500 hens/ha in all the flocks. Hen monitoring was conducted using Radio Frequency Identification (RFID) leg bands (Monza R6 UHF RFID Tags, Impinj, Seattle, WA, USA) and custom-built RFID antennae [35]. The shed interior of all sheds was equipped with two three-tier aviary system using chain-feeders and perches, with nest boxes being placed in the middle tier (Natura Step, Big Dutchman, MI, USA). Hen movement in this system was monitored (from 4 am to 8 pm) by placing RFID antennae 30 cm apart along the right and left side of the feeder line at the top tier (referred to as “upper feeder tier”) and bottom tier (referred to as “lower feeder tier”), as well as placing one RFID antenna at the entrance of the nest box, located on the middle tier. Further details regarding the monitoring system can be found in Sibanda et al. (2020) [35]. All experimental hens were individually weighed before range access was first provided at 16 weeks of age and weighed again at 21 weeks of age (Veit BAT 1, Moravany, Czech Republic). From the age of 21 weeks onwards, the experimental hens were partitioned according to their range use into groups consisting of “stayers” (hens that preferred to stay in the shed on most days) or “rangers” (hens that accessed the range on most days) and continued to be monitored for their range and aviary use until 74 weeks of age. All hens had equal access to all resources (the aviary system, both feeders, drinkers, nest boxes, and the range) but could not leave their partition (e.g., stayers or rangers) due to the trans-sectional divisions in the hen house and the 1.8–1.2 m high fence that separated the ranges. Equal numbers of stayers (*n* = 625) and rangers (*n* = 625) were placed in each partition, allowing for a comparable stocking density. Figure 1 outlines the study population: Hens were allowed an acclimatization period of 2 weeks to the range (16–18 weeks of age), subsequently all hens were monitored for their range and aviary use between 18 and 21 weeks of age. All three flocks had the same arrangement inside the shed as well as structure and orientation of the shed and were carried out in parallel in different sheds (few weeks apart to allow ease of sampling).

At 74 weeks of age, total of 894 hens (targeting around 300 hens (150 stayers and 150 rangers)/flock; flock 1: *n* = 274, flock 2: *n* = 341, flock 3: *n* = 279) from all three flocks were sub sampled from the 3125 hens/flock. The experiment was carried out in three flocks separately when the flocks reached their final age (74 weeks). For all the experiments in each flock, hens were randomly selected from both the stayer and ranger pens for the current study. Random selection was achieved by allocating various locations (*n* = 34) in each pen an arbitrary number. The locations were chosen to reflect each level of the aviary system (Figure 2A), distance from pop holes (Figure 2B) and distance from nearest fence, wall, feeder, and open area. Hens were collected from the locations in a specific order determined by a random sequence generator (https://www.random.org/sequences/). Individual hens from each location were visually selected from approximately one-meter distance and caught within 30 s; ensuring that the most fearful birds were not excluded due to flight distance responses when approached and thus minimized bias due to ease of capture but also ensuring that hens were not chased for extended periods increasing stress prior to the test. The hen was immediately placed in a bucket closed with a lid that allowed for minimal vision but adequate air circulation. All hens were caught and placed in the test arena within 5–7 min. When hens were not caught within 30 s, the researcher moved out of the area and waited until the hens were back in that area repeating the process until a hen was caught. The behavior testing lasted 5 days for each flock, allowing 60 hens/day to be evaluated in four arenas (20 hens/arena/day).

While initially 894 hens were collected and tested for behavior analysis (targeting 300 hens/flock), 125 hens were excluded from the statistical analysis due to malfunctioning RFID tags, incomplete data sets and testing errors. This resulted in a total study population of 769 hens including 404 stayers and 365 rangers (flock 1: *n* = 176, flock 2: *n* = 323, flock 3: *n* = 270). Details of these 769 hens describing their range and aviary use are presented in Table 1 and Table 2.

### 2.2. Behavioral Tests

The selected hens were subject to two behavioral tests to assess fearfulness and curiosity/exploration: The Novel Arena (NA) test and Novel Object (NO) test. Four test arenas (17 m^2^) constructed of CD plywood (Ecoply, Box Hill, VIC, Australia) with floor covering wood shavings (Breeders choice, Colac, VIC, Australia) were used to conduct the test. Each arena was placed approximately 100 m from the hen house and approx. 20 m from each other in an open spaced area and ensonified with white noise (speakers: Huizhou Lpar Technology Co., Ltd., Huizhou, Guangdong, China) to minimise the interference with any potential external sounds during the testing procedure. Each arena was equipped with an overhead video camera (Sony HDRCX625 Full HD Handycam, Sony corporation, Beijing, China) which allowed for continuous visual and sound recording and data analysis at a later timepoint.

For the NA test, hens were placed in the center of the arena, facing away from the researcher, and then left for 8 min with the door of the arena closed [11]. Immediately thereafter, the NO test was conducted where a novel object (dog play rope of mixed color 20 cm in length, taped to a neon-pink plastic colored box 26.7 × 18.3 cm; Figure 3) was introduced within 3 s through a small trap door (20 × 26.5 cm) on the back wall of the arena, minimizing human contact [11]. The NO was left with the hen in the arena for 5 min. After the completion of the NO test, the number of defecations were counted and removed from the bedding material. Each hen was then weighed using poultry weighing scales (Veit BAT 1, Moravany, Czech Republic) and returned to the flock.

### 2.3. Validation of ANYmaze Software Data

To our knowledge, hen behavior had not previously been assessed using ANYmaze software. Therefore, to validate the system, the time spent in NO interaction zone, NO approach zone, and NO avoidance zones as well as the number of lines crossed was recorded by one observer manually for 25 randomly selected hens and compared to the results obtained for these same hens generated by the ANYmaze software. To assess whether the ANYmaze software and manual method agreed significantly, the Bland–Altman method of limits of agreement was used [36]. The bias (average difference) between the two methods (manual over ANYmaze) where manual recorded a 0.01%, 3.12%, −4.52%, and −6.92% deviation from the software’s recordings for time spent in NO approach zone, interaction zone and avoidance zone and the number of lines crossed during the NA test, respectively. The results from Bland–Altman method indicated that the two methods were in acceptable agreement for all parameters investigated, except for parameter number of lines crossed (Suppl1). The difference between the manual and ANYmaze for the number of lines crossed was most likely because the software considered lines crossed differently than the observer: The observer considered lines crossed when 1/2 of the hen’s body had moved over the relevant line, while the software considered it when ¾ of the hen’s body had moved over the line, applying a more conservative approach. Therefore, data from ANYmaze was used for further analysis of the study population.

### 2.4. Data Processing

ANYmaze software (Stoelting Co., Wood Dale, IL, USA) was used to automatically collect behavioral data from the video. For the NA test, grid lines were prepared and overlaid the video footage virtually and divided the recorded floor arena in 16 equal squares (0.106 m^2^ each) (Figure 4A). The latency until the first step, the number of lines crossed, the total distance covered, the number of vocalization and the number of escape attempts were assessed as indicators of fear. For the NO test, the arena was divided into 4 zones of interest (Figure 4B): The NO interaction zone (within 10 cm radius of the NO), the NO approach zone (area within 40 cm radius of the NO), and the NO avoidance zone (>40 cm area distance from the NO). For the NO test, the total distance covered, time spent in the NO interaction zone, time spent in NO approach zone and time spent in the NO avoidance zone was determined. Time spent in different areas during the NO test, number of vocalizations (all individual vocalizations, irrespective of type was recorded) and the number of escape attempts were assessed as indicators of neophobia and exploration during the NO test. The software was unable to recognize latency to first step (NA test), number of vocalizations (NA and NO test), and escape attempts (NA and NO test), therefore these data were manually recorded by one observer blind to treatment.

We defined the age between 18–21 weeks of ranging as early life range use and the age between 18–74 weeks of ranging as whole life range use. At 74 weeks of age (at end of lay), with NA and NO tests, hens’ fearfulness and exploration were assessed and its relationship with the outdoor range use and location inside the shed were determined conducting the following statistical analysis.

### 2.5. Statistical Analysis

#### 2.5.1. Relationship between Early Life Range Use (18–21 Weeks of Age) and Whole Life Range Use (18–74 Weeks of Age)

The relationship between early (18–21 weeks of age) and whole life (18–74 weeks of age) range use/hen and mean duration per visit/hen were analyzed using linear regression in JMP 14 statistical analysis software (SAS Institute Inc., Cary, NC, USA). The data for average time spent on the range, upper feeder tier, lower feeder tier and nest box tier did not meet the criteria for normality and was therefore square root transformed.

#### 2.5.2. Pearson’s Correlation and Random Forest Analysis to Determine the Most Important Predictors for Early Life and Whole Life Range Use

The variables were tested for multicollinearity using Pearson’s correlation using corrplot package [37] in R studio [38] and removed from analysis to produce robust and accurate model as highly correlated variables were found to affect the Random Forest Model performance, accuracy and the order of least important variables [39,40,41]. In detail, the variables that had correlation values greater than 0.70 were identified and one variable was removed from the analysis. Time spent in the NO approach zone was correlated with the latency to first step in the NO approach zone (r = −0.71) and the time spent in the NO interaction zone was correlated with the latency to first step in the NO interaction zone (r = −0.72). Therefore, latency to first step in the NO approach zone and NO interaction zone were removed from analysis. Similarly, the body weight at 74 weeks was removed from the analysis due to high correlation with body weight gain at week 16–74 (Suppl2). The final ranging variables and behavior assessment variables (3 body weight indicators and 13 fearfulness/curiosity parameters; Table 3) were subjected to Random Forest Analysis to identify the important variables to explain use of the aviary system, to minimize the impact of unimportant variables on the power of the model and increase accuracy [41,42]. Individual hens were the experimental unit for the analysis.

#### 2.5.3. Model Training and Test Dataset

The variable importance analysis was run to determine the mean decrease accuracy (%IncMSE) values during the construction of Random Forest Model and the ranking of the variables differed according to the response variables (Suppl3, Suppl4, Suppl5; [43]). Before running the random forest model, model training and test dataset preparation was carried out. For the purpose of training the Random Forest, training data were set (80% of data) from the original data using the Random Forest package in R [43]. For the optimization of mtry value (number of variables available for splitting at each tree node), the caret package in R environment was used with default ntree (number of decision tress to grow) value of 500 with repeated 10-fold cross-validation technique for better estimation of performance, prevention of overfitting and improving the accuracy by using the same data for training and testing the classifier [40]. After obtaining the best mtry value, the combination of ntree and mtry was used to determine %IncMSE value for the test dataset. With %IncMSE reflecting the outcome of the cross-bag test, it was the preferred parameter to determine the relevant features compared to the IncNodePurity result, which is a training by-product. The magnitude of the %IncMSE value indicated the importance of the feature for the model. After running the first round of Random Forest Model training and obtaining the %IncMSE value of each variables from all 16 variables, the Random Forest Model performance was assessed by running forward feature selection and verifying the impact of variables inclusion over the model’s Root Mean Square Error (RMSE) values. An iterative approach was followed to determine at the lowest RMSE values of each combination (top predictors were selected one by one and the random forest model was run) to identify the point where increasing the number of variables was no longer an improvement to the model’s accuracy (Table 4). This process of variable inclusion assured the identification and removal of irrelevant variables from the model that may have mislead the algorithm and increased error. Based on this analysis, we selected the top important predictors that optimized the model performance (combination to give lowest RMSE values for each outcome variables; Table 3) and ran the Generalized Linear Mixed Model (GLIMM) analysis to determine the association between these predictors and the response variables.

#### 2.5.4. Relationships between Parameters of Fearfulness/Curiosity, Body Weight, and Use of the Aviary System

Relationships between the selected features of fear/curious behavior, body weight and different ranging variables (average time on the range) and location in the aviary system (lower and upper feeder tier and nest box tier) were analyzed using GLIMM with Gaussian distribution. Each model included the top predictor variables obtained from Random Forest as fixed factors and flock as a random factor. SPSS statistical software (v25, IBM Crop, Armonk, NY, USA) were used for GLIMM analyses.

## 3. Results

### 3.1. Description of Study Population

A low percentage of hens never accessed the range during their early life (18–21 weeks of age) (Flock 1: 11.8%, *n* = 91; Flock 2: 18.2%, *n* = 140, Flock 3: 8.19%, *n* = 63) or during whole life (18–74 weeks of age) (Flock 1: 1.30%, *n* = 10; Flock 2: 1.43%, *n* = 11; Flock 3: 0.26%, *n* = 2; Table 3). A wide variation of hen responses during the behavioral tests were observed, for example: 28 (*n* = 3.64%) and 37 (*n* = 4.81%) hens did not move at all during the NA and NO test respectively, whilst 345 (44.9%) hens stepped within 10 s (Table 3).

### 3.2. Relationship between Range Use between 18–21 and 18–74 Weeks of Age

Early life range use was significantly, but poorly correlated with whole life range use, including the total days of range use (R^2^ < 0.42), the total duration of range use (R^2^ ≤ 0.20) and mean duration per visit (R^2^ ≤ 0.20) (all *p* < 0.05; Suppl6, Suppl7, Suppl8).

### 3.3. The Most Important Predictors for Space Use Determined by Random Forest Regression Model

Table 4 presents the final variables selected for the Random Forest Analysis and used for the prediction of ranging and aviary location.

### 3.4. Relationships between Body Weight and Behavior Testing

Heavier pre-ranging body weight (at 16 weeks) was associated with more lines crossed during the NA test (F_(1,751)_ = 4.04, *p* = 0.045). There was no significant association between Δ body weight during 16–21 weeks of age or Δ body weight during 16–74 weeks of age with NA or NO behaviors (all *p* > 0.05).

### 3.5. Relationships between Use of Space between Early Life and NO and NA Behavior at 74 Weeks of Age

More time spent on the range during early life (16–21 weeks of age) was associated with more escape attempts during the NA test (F_(1,752)_ = 5.25, *p* = 0.0001; Table 4), heavier pre-ranging body weight (F_(1,752)_ = 62.0, *p* = 0.0001; Table 4) and greater early life body weight gain (F_(1,752)_ = 53.2, *p* = 0.0001; Table 4). More time on the upper feeder tier during early life was associated with more escape attempts during NA test (F_(1,753)_ = 2.25, *p* = 0.037; Table 4), lower body weight at 16 weeks of age (F_(1,753)_ = 52.4, *p* = 0.0001; Table 4) and less early life body weight gain (F_(1,753)_ = 59.9, *p* = 0.0001; Table 4). Conversely, more time spent at the lower feeder tier during early life was associated with heavier body weight at 16 weeks of age (F_(1,753)_ = 38.1, *p* = 0.0001; Table 4), greater early life body weight gain (F_(1,753)_ = 48.5, *p* = 0.0001; Table 4), lower whole life body weight gain (F_(1,753)_ = 8.51, *p* = 0.004; Table 4), more vocalizations during the NO test (F_(1,753)_ = 5.06, *p* = 0.025; Table 4) and more lines crossed during the NA test (F_(1,753)_ = 3.98, *p* = 0.046; Table 4). More time spent at the nest box during early life was associated with heavier body weight at 16 weeks of age (F_(1,764)_ = 23.0, *p* = 0.0001; Table 4) and higher body weight gain during early life period (F_(1,764)_ = 69.7, *p* = 0.0001; Table 4).

There was no statistical effect of flock on time spent in various areas during early life (average time spent on the range/day: *p* = 0.370, time spent at the lower feeder tier: *p* = 0.489, time spent at the upper feeder tier: *p* = 0.432 and time spent at the nest box tier: *p* = 0.332).

### 3.6. Relationships between Use of Space between Whole Life and NO and NA Behavior at 74 Weeks of Age

More time spent on the range during whole life showed a trend of shorter latency to first step (F_(1,757)_ = 3.58, *p* = 0.06; Table 4), more vocalizations (F_(1,757)_ = 10.2, *p* = 0.001; Table 4) during the NA test and more time spent in the interaction zone during the NO test (F_(1,757)_ = 5.06, *p* = 0.025; Table 4).

More time spent on the upper feeder tier during whole life was associated with less time spent in the NO interaction zone during the NO test (F_(1,758)_ = 6.26, *p* = 0.013; Table 4) and fewer lines crossed (F_(1,758)_ = 12.1, *p* = 0.001; Table 4) during the NA test, and lower body weight at 16 weeks of age (F_(1,758)_ = 15.6, *p* = 0.0001; Table 4). Conversely, more time spent at the lower feeder tier during whole life was associated with heavier pre-ranging body weight (F_(1,750)_ = 10, *p* = 0.001; Table 4), more time spent on the NO interaction zone during the NO test (F_(1,750)_ = 8.71, *p* = 0.003; Table 4), more lines crossed (F_(1,750)_ = 7.35, *p* = 0.007; Table 4) and vocalizations (F_(1,750)_ = 14.3, *p* = 0.001; Table 4) during the NA test. More time spent at the nest box tier overall (whole life) was associated with the lower pre-ranging body weight (F_(1,754)_ = 5.57, *p* = 0.019; Table 4).

There was no effect of flock on time spent in various areas during entire ranging period (whole life, e.g., minutes spent on the range: *p* = 0.325; time spent at the lower feeder tier: *p* = 0.361; time spent at the upper feeder tier: *p* = 0.343; time spent at the nest box tier: *p* = 0.321).

## 4. Discussion

The hens in this study were assessed for their location preference within a free-range aviary housing system and we examined if location preferences were related to fearfulness and curiosity. We found little evidence that location was related to fearfulness, however we did find that hens that spent more time on the range and on the lower feeder tier for their whole life were more curious at the end of the production cycle.

The relationship between indicators of hen curiosity and range use was present only when we included their whole life ranging behavior, not just early life ranging. We suggest that these results indicate curiosity of these hens developed over time. The physical structure and complexity inside the shed (for example, nest boxes, bedding material, tiers and perches) and the range environment provides stimuli that may affect brain development (hippocampal neurogenesis), exploration, curiosity and fearfulness [34,44,45,46]. Indeed, our results suggest that a hen’s experience on the range and the lower feeder environments may have an impact on their temperament, perhaps through the novelty of the range environment, or the high-traffic area inside the shed at the lower tier. The exploration and range use of hens increased over time with repeated exposure and increased familiarity [29,30] suggesting possible effect of age and early life experiences [31]. Whilst we found no evidence that fearfulness was associated with time spent anywhere in the free-range aviary system, the number of hens that accessed the range slightly increased over time (from 88.7% to 98.7%, 81.8% to 98.6%, and 91.8% to 99.7% in flock 1, 2 and 3 respectively). This is in agreement with previous research that showed exploration and range use of hens increased over time with repeated exposure and increased familiarity [29,30] suggesting a possible effect of age and early life experience [31].

However, the increase in range use over time may also indicate that hens were fearful during the first few weeks of range access and fearfulness decreased over time. Alternatively, increased range use over time may have been a result of social facilitation as observed in broiler chickens [47]. We provide some evidence that curiosity is related to use of the range and lower tier environment but cannot determine causation, this work furthers the understanding of this growing field and generates hypotheses for further investigation.

Neither pre-ranging body weight, nor body weight change during early life ranging was associated with any indicators of fear or curiosity at the end of the production cycle. This rejects our hypothesis that fearful birds would be lighter in body weight. However, pre-ranging body weight and body weight gain during early life ranging were the most important factors in predicting time spent on the range and at the lower feeder tier during early life ranging. Heavier hens spent more time on the range and at the lower feeder tier and lighter hens spent more time at the upper feeder tier. This is in contrast to a study in broilers, where range use was associated with a reduction in body weight gain of the birds, which was thought to be related due to increased thermoregulation, stress responses, level of activity, or a combination of factors [23]. While the same factors might have affected the layers in the present study, their increased maturity at 16 weeks of age including the presence of a solid feather cover might have minimized the impact on energy maintenance [48,49]. We propose that the hens that preferred the upper feeder tier may have been lower in the social rank as reflected by their lower body weight motivating these lighter hens to seek refuge in higher spaces. This behavior has been previously demonstrated where low-ranking hens spent more time on a perch that served as a refuge [25,50]. Several other studies have shown that priority of access to resources is positively associated with social ranking, which might have prevented the hens in the present study accessing the lower feeder tier or the range [25,51]. Gibson et al. (1986) [52] found that hens of lower social rank had restricted areas of movement as well as reduced body weight and poor feather cover providing evidence that social factors influence the way individual hens move and use available resources. Hens in large groups replace the hierarchical (pecking order) social system with a more anonymous social ranking system that depends on body and comb size [25,53,54,55]. Having 625 hens present in each partition, this anonymous social rank system would likely to apply in the present research study, as it has been suggested that hens can only recognize ≤ 120 individual flock companions [56]. Therefore, it is possible that the hens on the upper feeder tier in this study might have been of lower social rank and chose to stay in the upper feeder tier to avoid inter-hen aggression. The function of a “refuge” space at the upper feeder tier may therefore be an essential feature in the hen house to reduce stress and may even decrease the potential fearfulness associated with social dominance/structure/hierarchy. Although, we did not find evidence that fear, hen location and/or body weight were associated to each other, the hypothesis of an upper tier refuge as well as the relationship to body weight is worth further investigation. This might help in designing the commercial hen house and optimize uniform body weight during rearing to safeguard hen welfare.

We used the NA test to evaluate fearfulness where we interpret a passive response (freezing, fewer vocalization, longer latencies) as indicative of fearfulness and a higher level of activity (shorter latencies, more lines crossed) indicative of decreased fearfulness [11,57,58]. Moreover, we consider that the active responses in the NA test may be interpreted as a heightened motivation for social reinstatement or a proactive response to fear provoking stimuli [11].While previous studies have found indoor hens to be more fearful than the range users [2,9,59], the findings of the present study showed that there was no relationship of range use and fearfulness, of note the aforementioned studies were all conducted in controlled research conditions. Larsen et al. (2018) [3] is the only other study to investigate these relationships in commercial conditions, and found ranging hens were more likely to approach a NO (reduced neophobia, or increased curiosity) in line with our findings. But again, the findings from this study in relation to fearfulness were inconsistent during early life and whole life as well as within each timepoint and therefore, strong conclusion could not be made.

More time spent at the upper feeder tier was associated with less time spent in the NO object interaction zone, which may be reflective of neophobia. However, as these hens showed no freezing or avoidance behavior both during early life and whole life, the reluctance to approach the NO may rather be reflective of less curious temperaments. Time spent at the lower feeder tier was associated with more time in the NO interaction zone as well as more vocalization during the NA test which we interpret as increased curiosity and social motivation. However, Sibanda et al. (2020) [4] showed evidence of the strong relationship between time spent at the lower feeder tier and time spent on the range suggesting that these hens may have been more likely to visit the range environment. Perhaps the relationships between lower feeder tier use and curiosity was related to their use of the range area rather than the lower feeder environment per se.

As we found little evidence that fearfulness was related to time spent at any location during early or whole life, we considered whether coping styles in response to fear provoking stimuli was associated with time spent at each location rather than the degree of fearfulness. Coping styles of animals are commonly defined by their response to a stimulus that is either active/proactive (fight or flight) or passive/reactive (freezing) [59,60]. Proactive individuals have been described as maintaining rigid inflexible routine behavior [60,61,62,63]. Campbell et al. (2016) [2] suggested that proactive coping style was associated with laying hens that used the range, however, found no strong evidence of relationships between range use and behavioral indicators of coping style. In agreement, we found no evidence that early life range use or whole life range use was associated with a specific coping style.

This study is the most comprehensive report regarding the relationship of curiosity, fear and coping styles with horizontal and vertical space use as well as body weight of commercial free-range laying hens to date. Although we provide some hypothesis-generating evidence, we cannot determine causation. Furthermore, because hens were individually handled at 16 and 18 weeks of age, and also grouped in separate pens, we cannot exclude the fact that the social facilitation may have impacted the hen behavior in various ways.

## 5. Conclusions

This study aimed to identify relationships between time spent in aviary locations during early ranging and whole life ranging and indicators of fearfulness and curiosity in commercial free-range hens. We found no evidence that hen preferences for locations within the housing system were associated with fearfulness, during early or whole life range access periods. Rather we provide evidence that time spent on the range and at the lower feeding tier was associated with curiosity. Moreover, it is likely that curiosity increased over time, suggesting that environmental stimuli could be used strategically to raise more curious hens. Further work is required to better understand the complex relationships between dominance rank, body weight, sociality, curiosity, and use of space (including ranging behavior) to optimize commercial hen house design to safeguard hen welfare.

## Figures and Tables

**Figure 1 animals-10-01706-f001:**
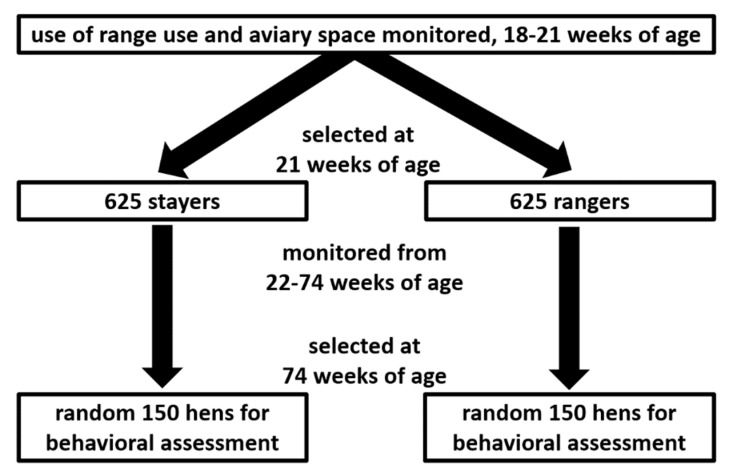
Flow chart representing the selection process of experimental hens subject to the behavior tests in each of the three flocks investigated on a commercial farm. While 300 hens/flock were targeted and tested for behavior analysis, 769 hens were included in the statistical analysis due to incomplete datasets. The 625 stayers and rangers present in each of the three flocks were housed in separate pens as required for a parallel experiment.

**Figure 2 animals-10-01706-f002:**
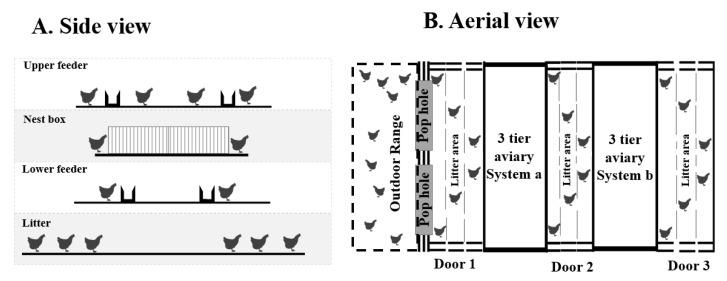
Pictorial diagram of locations that hens were randomly selected from within the commercial 3-tiered aviary system. (**A**) represents a side view and (**B**) represents an aerial view of the aviary system showing the floor area with litter, lower feeder tier, nest box tier and the upper feeder tier as well as the outdoor range. Hens could freely access any area according to their choice, except when the pop holes to the range were closed during 8 pm until 4 am. For the random hen selection for behavior tests at 74 weeks of age, each location was allocated a random number and the order of locations were determined using a random sequence generator (https://www.random.org/sequences/) corresponding to the allocated location number. Thus, the randomly created sequence was used to select hens from all sections of the aviary. The pen had three entry doors (B. aerial view) which were entered sequentially (Door 1 first, Door 2 s, Door 3 third, Door 1 fourth, and so on) to collect two hens per visit according to the numbered sequence. This sequence was followed until the targeted number of hens was tested.

**Figure 3 animals-10-01706-f003:**
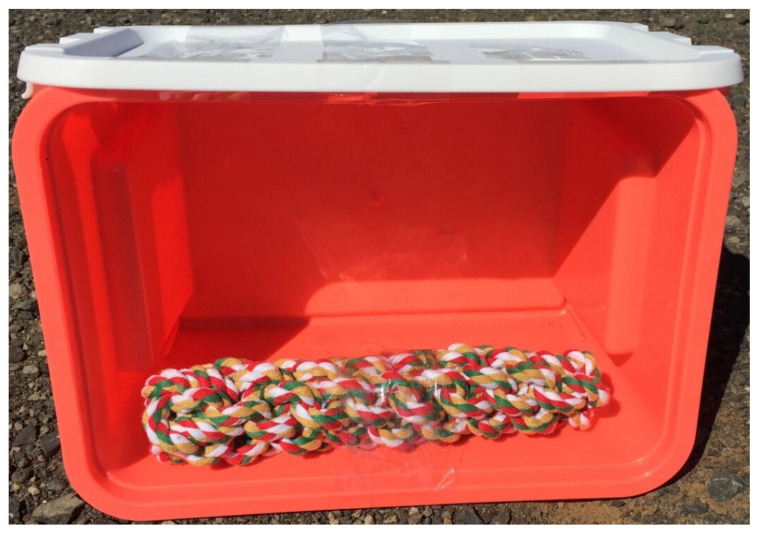
A dog play rope placed inside of a box was used for the Novel Object test and was fitted with a white lid on the top. The Novel Object was placed with the rope facing the hens.

**Figure 4 animals-10-01706-f004:**
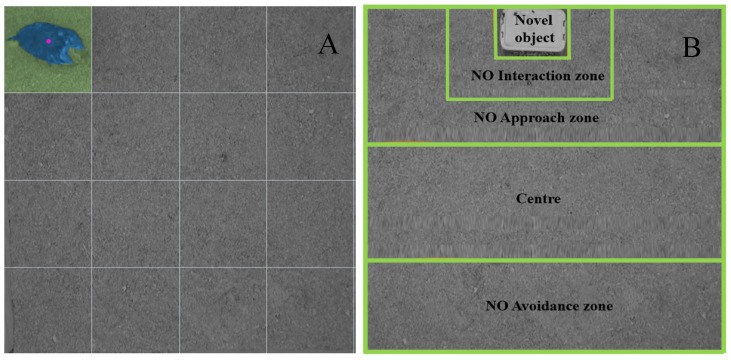
Layout of novel arena (NA) test from the ANYmaze software that was used to determine different parameters of NA test in the laying hens of a commercial farm. The novel arena was 1.7 m^2^ and the lines were virtually overlaid on the video footage using the ANYmaze software to divide the NA into 16 equal squares. (**A**) shows the tracking of the hens by ANYmaze software to determine the number of lines crossed and the total distance moved/hen in the NA. (**B**) Layout of novel object (NO) test from the ANYmaze software that was used to determine different parameters of NO test. The lines were virtually overlaid on the video footage using the ANYmaze software to divide the NA into four zones of interest; NO avoidance zone, Centre, NO approach zone and NO interaction zone. Automatic tracking at different zones was done using the software after placing the NO from the trap door. The bright green line across the figure is the virtual scale from the software overlaid on the video used for measurement and dividing the arena into zones. Time spent at these zones and total distance moved were some of the measured parameters.

**Table 1 animals-10-01706-t001:** Range use (mean ± SEM) of hens (*n* = 769) obtained from 3 flocks that were categorized and housed according to their range use during early (18–21 weeks of age) and whole (22–74 weeks of age) ranging life.

	Rangers			Stayers		
Flock	1	2	3	1	2	3
**18–21 weeks of age**						
Number of days that the range data was available	26	24	28	26	24	28
Average number of days the range was accessed	11.6 ± 0.68	15.4 ± 0.46	22.9 ± 0.47	0.50 ± 0.20	2.18 ± 0.41	3.08 ± 0.48
Average percentage of days the range was accessed	44.6 ± 2.61	64.29 ± 1.93	81.9 ± 1.67	1.94 ± 0.76	9.07 ± 1.72	10.9 ± 1.70
Average time spent on the range (minutes/hen/day)	36.2 ± 3.17	47.2 ± 2.10	68.6 ± 2.81	1.36 ± 0.62	6.05 ± 1.43	4.50 ± 1.19
**22–74 weeks of age**						
Number of days that the range data was available	344	330	293	344	330	293
Number of days that hens accessed the range	209.5 ± 5.70	233.5 ± 4.73	254.0 ± 2.27	112.6 ± 8.47	128.0 ± 8.12	149.8 ± 7.42
Percentage of days that hens accessed the range	60.9 ± 1.66	70.8 ± 1.43	86.7 ± 0.77	32.7 ± 2.46	38.8 ± 2.46	51.1 ± 2.53
Average time spent on the range (minutes/hen/day)	34.5 ± 1.78	26.5 ± 1.36	62.5 ± 2.02	19.4 ± 1.95	23.4 ± 2.16	44.0 ± 3.29

**Table 2 animals-10-01706-t002:** Time average spent at various location inside the shed (mean ± SEM) of hens (*n* = 769) obtained from 3 flocks that were categorized and housed according to their range use during early (18–21 weeks of age) and whole (22–74 weeks of age) ranging life.

	Rangers			Stayers		
Flock	1	2	3	1	2	3
**18–21 weeks of age**						
Lower feeder tier (min/hen/day)	393.9 ± 16.7	470.1 ± 14.5	543.6 ± 19.1	218.9 ± 22.2	170.8 ± 18.8	262.7 ± 26.0
Upper feeder tier (min/hen/day)	117.1 ± 11.5	111.2 ± 8.52	135.9 ± 10.4	369.6 ± 20.8	440.3 ± 17.9	494.3 ± 23.5
Nest box tier (min/hen/day)	55.5 ± 3.46	55.9 ± 2.47	87.9 ± 3.77	48.4 ± 4.56	47.3 ± 3.09	74.7 ± 5.37
**22–74 weeks of age**						
Lower feeder tier (min/hen/day)	423.7 ± 19.1	310.2 ± 12.8	445.2 ± 17.8	311.9 ± 24.2	229.1 ± 18.7	275.2 ± 19.8
Upper feeder tier (min/hen/day)	130.0 ± 13.2	84.2 ± 6.46	125.5 ± 7.97	279.9 ± 17.6	197.0 ± 0.09	301.2 ± 16.6
Nest box tier (min/hen/day)	58.8 ± 3.05	39.9 ± 1.78	85.2 ± 4.81	71.2 ± 4.48	48.0 ± 2.22	106.4 ± 5.86

**Table 3 animals-10-01706-t003:** Descriptive statistics (stayers and rangers combined) illustrating mean ± SEM, minimum and maximum values for all the variables obtained from individual hens across 3 flocks on a commercial free-range farm.

Variables	Flock 1 (*n* = 176)		Flock 2 (*n* = 323)		Flock 3 (*n* = 270)		Total (*n* = 769)	
	Mean ± SEM	Min–Max	Mean ± SEM	Min–Max	Mean ± SEM	Min–Max	Mean ± SEM	Min–Max
Range use week 18–21 (min/day)	20.5 ± 2.32	0–147.5	26.8 ± 1.73	0.01–113.1	36.1 ± 2.50	0.02–194.0	28.6 ± 1.28	0.02–194.0
Range use week 18–74 (min/day)	25.1 ± 1.47	0–94.0	24.5 ± 1.27	0–128	52.7 ± 2.01	0–150	34.6 ± 1.06	0–150
Lower feeder tier use 18–21 (min/day)	289.5 ± 16.1	0.8–908.8	320.9 ± 14.5	0–940.8	402.1 ± 18.2	1–1107.4	342.2 ± 9.71	0–1107.4
Lower feeder tier use 18–74 (min/day)	356.6 ± 16.8	0–893	269.4 ± 11.5	0–1070	359.1 ± 14.3	4–1048	320.8 ± 8.11	0–1070
Upper feeder tier use 18–21 (min/day)	267.7 ± 16.2	0–953.7	275.3 ± 13.5	0–929.3	316.4 ± 16.9	0–1073.8	288.0 ± 9.92	0–1073.8
Upper feeder tier use 18–74 (min/day)	219.0 ± 13.0	0–800.0	140.0 ± 6.39	0–569	213.5 ± 10.7	0–828	183.9 ± 5.64	0–828
Nest box tier use 18–21 (min/day)	51.3 ± 3.06	0–216.2	51.6 ± 1.99	0–249.3	81.2 ± 3.31	0–364.7	62.0 ± 1.67	0–364.7
Nest box tier use 18–74 (min/day)	65.8 ± 2.97	0–288.0	43.5 ± 1.44	0–174	95.4 ± 3.84	7–398	66.8 ± 1.82	0–398
Body weight at week 16 (kg)	1.26 ± 0.00	0.8–1.5	1.26 ± 0.00	0.9–1.66	1.34 ± 0.00	1.01–1.7	1.29 ± 0.00	0.85–1.70
Δ body weight from week 16–21 (kg)	0.46 ± 0.00	−0.3–0.8	0.46 ± 0.3	−0.3–0.76	0.38 ± 0.01	−0.51–0.79	0.43 ± 0.01	−0.51–0.82
Δ body weight from week 16–74 (kg)	0.70 ± 0.01	−0.2–1.0	0.70 ± 0.00	0.02–1.71	0.60 ± 0.01	−0.1–1.55	0.63 ± 0.01	−0.18–1.71
Total distance moved during the NO test (m)	6.83 ± 0.56	0–45.4	5.47 ± 0.32	0–32.3	11.8 ± 0.84	0–111.1	8.01 ± 0.36	0–111.1
Time spent in the NO avoidance zone (s)	101.0 ± 7.89	0–300	118.4 ± 6.38	0–300	118.0 ± 6.55	0–300	114.3 ± 3.97	0–300
Time spent in the NO approach zone (s)	90.5 ± 6.71	0–300	70.8 ± 4.80	0–300	68.1 ± 4.78	0–300	74.3 ± 3.03	0–300
Time spent in the NO interaction zone (s)	25.4 ± 4.39	0–281.5	19.8 ± 2.86	0–294.4	15.1 ± 2.52	0–300	19.5 ± 1.80	0–300
Escape attempts/hen during the NO test	0.0 ± 0.01	0–1.0	0.05 ± 0.02	0–3.0	0.01 ± 0.01	0–1.0	0.03 ± 0.01	0–3.0
Escape attempts/hen during the NA test	0.5 ± 0.07	0–5.0	0.11 ± 0.02	0–3.0	0.34 ± 0.05	0–6.0	0.28 ± 0.03	0–6.0
Total distance/hen moved during the NA test (m)	15.0 ± 0.79	0–56.2	11.0 ± 0.51	0–50.0	13.5 ± 0.88	0–92.6	12.8 ± 0.42	0–92.6
Total number of lines crossed during the NA	45.8 ± 2.27	1–148	36.4 ± 1.52	0–114	41.1 ± 2.49	0–291	40.1 ± 1.21	0–291
Latency to first step during the NA test (s)	38.5 ± 5.36	1–480	62.8 ± 6.02	1–480	71.9 ± 6.91	0–480	60.4 ± 3.74	0–480
Vocalizations during the NA test/hen	3.0 ± 0.52	0–49	5.80 ± 0.54	0–74	6.45 ± 0.48	0–47	5.39 ± 0.31	0–74
Vocalizations during the NO test/hen	4.06 ± 0.61	0–46	5.43 ± 0.49	0–63	5.93 ± 0.46	0–45	5.29 ± 0.30	0–63
Total number of excreta droppings during the NA and NO test/hen	1.34 ± 0.08	0–6	1.12 ± 0.04	0–4	1.19 ± 0.06	0–6	1.20 ± 0.03	0–6

**Table 4 animals-10-01706-t004:** The root mean square error (RMSE) values (number of selected features in parentheses) obtained based on the confusion matrix to assess the model performance and to determine the number of features (behavior and body weights parameters) to be included in further analysis obtained from Random Forest Analysis. Generalized Linear Mixed Model Analysis determined the relationship between the predictor (time spent on the range, lower feeder tier, upper feeder tier, or nest box tier) and response variables to determine the relationship between use of space in an aviary housing system and behavioral indicators of fearfulness and exploration. β coefficient values indicate the direction and strength of the relationships and significant relationships are indicated by *p*-values of <0.005.

	Random Forest Analysis			Generalized Linear Mixed Model Analysis			
	RMSE	RMSE_FS_	Selected Variable	β Coefficient	CI Lower	CI Upper	*p*-Value
**Early life** **(18–21 weeks of age)**							
**Average range use (minutes/hen/day)**	15.7	17.2 (8)	Body weight at week 16	9.55	7.17	11.9	0.0001
			Δ Body weight from 16–21	6.46	4.72	8.20	0.0001
			Escape attempts during the NA test	−1.73	−8.52	5.07	0.618
			Latency to first step during the NA test	−0.001	−0.004	0.001	0.342
			Time spent in the NO avoidance zone	0.001	−0.002	0.004	0.399
			Time spent in the NO approach zone	0.002	−0.001	0.006	0.203
**Average time spent on the lower feeder tier (minutes/hen/day)**	116.3	114.9 (6)	Body weight at week 16	16.7	22.4	21.9	0.0001
			Δ Body weight from 16–21	13.8	9.92	17.7	0.0001
			Δ Body weight at week 74	−0.26	−1.90	1.38	0.753
			Δ Body weight from 16–74	−5.40	−9.03	−1.77	0.004
			Avoidance-Approach towards NO during NO test	−3.08	−19.5	13.3	0.713
			Number of lines crossed during the NA test	0.02	0.000	0.034	0.046
			Vocalizations during the NO test	0.08	0.01	0.145	0.025
			Total distance moved in the NO test	0.004	−0.22	0.03	0.769
**Average time spent on the upper feeder tier (minutes/hen/day)**	84.5	98.6 (8)	Body weight at week 16	−18.9	−24.1	−13.8	0.0001
			Δ Body weight from 16–21	−14.7	−18.4	−10.9	0.0001
			Escape attempts during the NA test	14.3	−0.896	29.4	0.065
			Latency to first step during the NA test	0.000	−0.005	0.006	0.945
			Time spent in the NO approach zone	0.001	−0.004	0.006	0.816
			Number of lines crossed during the NA test	−0.01	−0.027	0.008	0.266
			Vocalizations during the NO test	−0.42	−0.11	0.03	0.219
			Escape attempts during the NO test	−12.2	−27.3	2.82	0.111
**Average time spent on the nest box tier (minutes/hen/day)**	21.3	19.9 (4)	Body weight at week 16	4.61	2.73	6.50	0.0001
			Δ Body weight from 16–21	4.61	2.73	6.50	0.0001
			Latency to first step during the NA test	0.000	−0.002	0.002	0.767
			Time spent in the NO approach zone	0.000	−0.002	0.002	0.767
**Whole life** **(18–74 weeks of age)**							
**Average time spent on the range (minutes/hen/day)**	14.7	15.5 (7)	Latency to first step during the NA test	−0.002	−0.004	0.00007	0.060
			Time spent in the NO avoidance zone	0.001	−0.001	0.003	0.262
			Time spent in the NO approach zone	0.001	−0.002	0.004	0.490
			Avoidance-Approach towards NO during NO test	1.21	−5.51	4.50	0.842
			Vocalizations during the NO test	0.018	−0.006	0.043	0.145
			Time spent in the NO interaction zone	0.004	0.001	0.008	0.025
			Vocalizations during the NA test	0.04	0.02	0.06	0.001
**Average time spent on the lower feeder tier (minutes/hen/day)**	98.1	93.4 (9)	Body weight at week 16	7.34	2.88	11.8	0.001
			Δ Body weight from 16–74	0.698	−2.24	3.63	0.641
			Time spent in the NO avoidance zone	0.002	−0.004	0.007	0.487
			Time spent in the NO approach zone	0.004	−0.003	0.011	0.269
			Number of lines crossed during the NA test	0.019	0.005	0.034	0.007
			Time spent in the NO interaction zone	0.015	0.005	0.024	0.003
			Vocalizations during the NA test	0.105	0.05	0.16	0.0001
			Defecations during the NA and NO tests	0.64	−6.79	8.07	0.866
			Δ Body weight from 16–21	−5.75	−8.60	−2.89	0.0001
**Average time spent on the upper feeder tier (minutes/hen/day)**	65.5	64.1 (9)	Δ Body weight from 16–74	−0.038	−2.64	2.56	0.977
			Latency to first step during the NA test	0.001	−0.003	0.006	0.480
			Time spent in the NO avoidance zone	−0.002	−0.007	0.003	0.369
			Time spent in the NO approach zone	−0.004	−0.01	0.002	0.192
			Avoidance-Approach towards NO during NO test	−1.35	−15.0	12.3	0.847
			Total number of lines crossed during the NA test	−0.023	−0.035	−0.010	0.001
			Escape attempt in the NO test	−4.70	−15.8	6.44	0.408
			Time spent in the NO interaction zone	−0.011	−0.019	−0.002	0.013
**Average time spent on the nest box tier (minutes/hen/day)**	21.2	19.9 (8)	Body weight at week 16	−1.90	−3.48	−0.32	0.019
			Δ Body weight from 16–21	0.13	−1.02	1.28	0.826
			Δ Body weight from 16–74	0.02	−1.05	1.09	0.969
			Latency to first step during the NA test	0.000	−0.002	0.001	0.805
			Time spent in the NO approach zone	0.001	−0.001	0.002	0.297
			Total number of lines crossed during the NA test	−0.004	−0.009	0.001	0.150
			Total distance moved in the NO test	0.001	−0.02	0.02	0.938
			Defecations	0.358	−2.27	2.99	0.789
**Body weight (kg/hen)**							
**Body weight at week 16 (kg/hen)**	0.056	0.056 (7)	Escape attempt in the NA test	−0.12	−0.33	0.09	0.281
			Time spent in the NO avoidance zone	−0.00001	0.000	0.00007	0.793
			Time spent in the NO approach zone	0.00001	0.000	0.000	0.852
			Total number of lines crossed during the NA test	0.000	0.00005	0.000	0.045
			Total distance moved in the NO test	0.00008	−0.001	0.001	0.830
			Time spent in the NO interaction zone	0.00003	0.000	0.000	0.666
			Defecations	−0.12	0.025	0.01	0.071
**Δ body weight between 16–21 weeks of age (kg/hen)**	0.09	0.083 (9)	Escape attempt in the NA test	0.14	−0.15	0.43	0.353
			Time spent in the NO avoidance zone	0.000	−0.00001	0.000	0.074
			Time spent in the NO approach zone	0.00008	−0.00007	0.000	0.315
			Total number of lines crossed during the NA test	0.000	−0.00008	0.001	0.147
			Vocalization during the NO test	0.001	0.000	0.002	0.187
			Total distance moved in the NO test	−0.001	−0.002	0.000	0.265
			Escape attempt in the NO test	−0.13	−0.42	0.16	0.390
			Time spent in the NO interaction zone	0.000	−0.00006	0.000	0.164
			Vocalization during the NA test	0.002	−0.002	0.002	0.411
**Δ body weight between 16–74 weeks of age (kg/hen)**	0.08	0.073 (4)	Escape attempt in the NA test	−0.007	−0.33	0.31	0.964
			Time spent in the NO approach zone	0.00003	0.000	0.000	0.805
			Total number of lines crossed during the NA test	0.00004	0.000	0.000	0.805
			Total distance moved in the NO test	−0.001	−0.002	0.001	0.278

Note: The abbreviations in the table represents: RMSE: Root mean square error, RMSE_FS_: Root mean square after feature selection, CI Lower: Lower Confidence interval, CI Upper: Upper Confidence interval.

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
