# Peer review of "Frequent Visits to an Outdoor Range and Lower Areas of an Aviary System Is Related to Curiosity in Commercial Free-Range Laying Hens"

_animals, 2020, doi:10.3390/ani10091706_

Round 1

Reviewer 1 Report

All my comments have been sufficiently addressed. I have no further comments. In my opinion this study helps us to understand the behavior of free-ranging laying hens.

Author Response

Thank you very much for taking your time to review this manuscript and helping to improve the quality.

Reviewer 2 Report

I do not know if it is a specific demand for the submission system. However, it is complicated to read the manuscript with all the changes (and different colors); the next time, if possible, please provide a clean version of the manuscript.

I read the authors' responses to the two reviewers and me. Most of the points were answered and clarified. However, some concerns remain.

From my point of view, the manuscript was easier to read in its previous version, even though it needed some modifications. Paragraphs, mainly in the discussion, are extremely long and difficult to follow. I did not understand the second part of the hypothesis (…conversely, relationships with choice of space, and whole production life ranging would indicate environmental influences on temperament).

In my opinion, in the title, it is preferable to state that these variables are related (like your previous title).

L26-29, L138-141, L453-456, and elsewhere – I am not sure we can say this.  In the first place, and as you mentioned, you tested your individuals just one time. Second, what if the individual temperament (exploration, curiosity…) varies during individual development, independent of their experience with multiple environments (although there may be correlations when we test it)? That is what happens with the behavioral responses of red junglefowl in a novel arena test, predator test, and a tonic immobility test (see Favati et al., 2016). I find the second part of your hypothesis very confusing and with very few supports in your text to better understand it since personality/temperament can change even in extremely controlled situations (see Bierbach et al., 2017). Why don't you just state that you want to check whether and how individual variation in later life (tested at the end of the production cycle) is related to behavior during early/whole life? It is simple, and you do not need to dive into complicated personality x environment influence questions…

 You should mention the measures and stocking density both in the barn and in the range.

L85 - Some words are lacking?

L119 – the use

L172-173 – acclimatization to the range, I suppose

L77- a total of

L293 – Maybe state that you recorded all individual vocalizations, independent of type? It is redundant to put it as individual vocalization since the tests were done individually.

L444-447 – I do not understand what this phrase is doing here. It does not fit as the most important results of your work, which is usually summarized in the first paragraph of a manuscript discussion.

L517-527 - Because you did find some relationships but their inconsistencies lead you to conclude that "the findings of the present study showed that there was no relationship of range use and fearfulness," I think it is useful here to further discuss (here, but also mention this in the abstract) the inconsistent findings of fearfulness between early and whole life periods.

467 – social facilitation

467-477 – Could you simplify this argument?

482 – indicators

493-496 – I'm not a native speaker, but this phrase seems odd.

Author Response

Thank you very much for your time in reviewing the manuscript.

Comment 1: I do not know if it is a specific demand for the submission system. However, it is complicated to read the manuscript with all the changes (and different colors); the next time, if possible, please provide a clean version of the manuscript.

Answer: We appreciate all the reviewer’s effort in helping to increase the clarity of the manuscript. We are sorry about the track changes in the manuscript, however, this is the requirement for the submission and we still need to include track changes while submitting the manuscript. We had sought an advice from the editor and they requested the manuscript with the track changes. For ease of reading the track changed manuscript, if we select the option “no Markup” in the review section, the changes will hide and make the manuscript more readable. We are sorry for the inconvenience.

Comment 2: I read the authors' responses to the two reviewers and me. Most of the points were answered and clarified. However, some concerns remain. From my point of view, the manuscript was easier to read in its previous version, even though it needed some modifications. Paragraphs, mainly in the discussion, are extremely long and difficult to follow.

Answer 2: Thank you very much for your suggestion. We have shortened the paragraphs as well as the sentences in various sections of discussion such as line: 467-481, 482-489, 513-519 to make it easy to follow.

Comment 3: In my opinion, in the title, it is preferable to state that these variables are related (like your previous title).

Answer 3: The title has been modified as “Frequent visits to an outdoor range and lower areas of an aviary system is related to curiosity in commercial free-range laying hens”.

Comment 4: I did not understand the second part of the hypothesis (…conversely, relationships with choice of space, and whole production life ranging would indicate environmental influences on temperament). L26-29, L138-141, L453-456, and elsewhere – I am not sure we can say this.  In the first place, and as you mentioned, you tested your individuals just one time. Second, what if the individual temperament (exploration, curiosity…) varies during individual development, independent of their experience with multiple environments (although there may be correlations when we test it)? That is what happens with the behavioral responses of red junglefowl in a novel arena test, predator test, and a tonic immobility test (see Favati et al., 2016). I find the second part of your hypothesis very confusing and with very few supports in your text to better understand it since personality/temperament can change even in extremely controlled situations (see Bierbach et al., 2017). Why don't you just state that you want to check whether and how individual variation in later life (tested at the end of the production cycle) is related to behavior during early/whole life? It is simple, and you do not need to dive into complicated personality x environment influence questions…

Answer 4:

We agree with reviewers comment that there is a complex relationship between personality and environment and we do not have a strong support from this experiment to clearly conclude about the impact of environment on the personality of the ranging hens. And, we agree that it can be argued/explained in both ways. This study being a longitudinal study, we have mentioned that we have provided some hypothesis generating evidence but could not determine the causation (line 552-553). Therefore, with the support of literature where it has been suggested that the complexity of environment (range, perches and ramps in aviary system) impacts the hippocampal neurogenesis, exploration and curiosity which might impact the range use. Since in this experiment we have provided hens opportunity of both range access and aviary system and the results obtained showing increase in the range use with time, we are more convinced that these might have impacted their range use. However, we understand the limitations of the study and discussed them in various places in the manuscript such as line: 477-480, line: 527-528, line: 550-555.

Comment 5: You should mention the measures and stocking density both in the barn and in the range.

Answer 5: Thank you very much for the suggestion. The information has been added in line 152-153 and it reads as follows:

“The indoor stocking density was 9 hens/m2 and outdoor stocking density was 1,500 hens/ha in all the flocks.”

Comment 6: L85 - Some words are lacking?

Answer 6: Thank you for your comment. We have added few words to increase the clarity of the sentence. The sentence reads as follows (Line: 87-91):

“In experimental and commercial housing systems consisting hens of 37-38 and 41 weeks of age [2, 18], behavioural assessments have shown that hens that prefer to stay inside the shed are generally more fearful than the hens that use the range frequently, as evident by more freezing behaviour and less movement in open field tests and longer tonic immobility durations [2, 18].”

Comment 7: L119 – the use

Answer 7: We have added ‘the’ (Line: 121).

Comment 8: L172-173 – acclimatization to the range, I suppose

Answer 8: Thank you very much. We have added ‘to the range’ and the sentence reads as follows (Line: 175-177):

Figure 1 outlines the study population: hens were allowed an acclimatization period of 2 weeks to the range (16 -18 weeks of age), subsequently all hens were monitored for their range and aviary use between 18 and 21 weeks of age.”

Comment 9: L77- a total of

Answer 9: Thank you very much for your comment. However, the sentence in line 77 has been removed from the manuscript.

Comment 10: L293 – Maybe state that you recorded all individual vocalizations, independent of type? It is redundant to put it as individual vocalization since the tests were done individually.

Answer 10: Thank you very much for the suggestion. We have added “all individual vocalisations, irrespective of type were recorded”.

The sentence reads as (Line: 296):

Time spent in different areas during the NO test, number of vocalisations (all individual vocalisations, irrespective of type were recorded) and the number of escape attempts were assessed as indicators of neophobia and exploration during the NO test.”

Comment 11: L444-447 – I do not understand what this phrase is doing here. It does not fit as the most important results of your work, which is usually summarized in the first paragraph of a manuscript discussion.

Answer 11: Thank you for your suggestion. The paragraph has been removed from the manuscript.

Comment 12: L517-527 - Because you did find some relationships but their inconsistencies lead you to conclude that "the findings of the present study showed that there was no relationship of range use and fearfulness," I think it is useful here to further discuss (here, but also mention this in the abstract) the inconsistent findings of fearfulness between early and whole life periods.

Answer 12: Thank you for the suggestion. Addition has been made both in the abstract (line: 42-43) and line 517-529 which reads as follows:

Line 42-43 (abstract): “The relationships during early and whole life use of space and some potential indicators of fearfulness were inconsistent and therefore, no strong, valid and reliable indicators of hen fearfulness such as freezing were identified.”

Line 527-529: “But again, the findings from this study in relation to fearfulness were inconsistent during early life and whole life as well as within each timepoint and therefore, strong conclusion could not be made.”

Comment 13: 467 – social facilitation

Answer 13: Thank you very much for your attention to details, the spelling has been corrected as: “social facilitation” (line 469).

Comment 14: 467-477 – Could you simplify this argument?

Answer 14: Thank you very much for your suggestion. The paragraph has been modified and reads as follows (line: 464-480):

“This is in agreement with previous research that showed exploration and range use of hens increased over time with repeated exposure and increased familiarity [29, 30] suggesting a possible effect of age and early life experience [31]. However the increase in range use over time may also indicate that hens were fearful during the first few weeks of range access and fearfulness decreased over time. Alternatively, increased range use over time may have been a result of social facilitation as observed in broiler chickens [47]. We provide some evidence that curiosity is related to use of the range and lower tier environment but cannot determine causation, this work furthers the understanding of this growing field and generates hypotheses for further investigation.”

Comment 15: 482 – indicators

Answer 15: Thank you for your attention to details. The spelling has been corrected (Line 482).

Comment 16: 493-496 – I'm not a native speaker, but this phrase seems odd.

Answer 16: The sentence has been rephrased and reads as follows (Line: 493-497):

“We propose that the hens that preferred the upper feeder tier may have been lower in the social rank as reflected by their lower body weight motivating these lighter hens to seek refuge in higher spaces.”

This manuscript is a resubmission of an earlier submission. The following is a list of the peer review reports and author responses from that submission.

Round 1

Reviewer 1 Report

I think this manuscript has merit, but due to grammar issues and the non-traditional analysis and presentation of data it was cumbersome to be able to assess the quality and interpret the scientific findings. 

Comments were made within the attached PDF,  did not comment about the results because the tables are very cumbersome and data was analyzed numerous ways--is there a better way to present data that is more meaningful to the scientific community?  

Maybe focusing on curiosity vs. fear and how these relate to welfare based on housing system may be more appropriate. 

Author Response

Thank you very much for the constructive comments to improve the manuscript. We have incorporated changes into the new draft of our manuscript indicated using track changes. Please find below an itemised list of the changes made in response to points raised.

Reviewer 2 Report

This paper aims to characterize two personality traits (exploration/curiosity and fearfulness) in laying hens and relate these to range and aviary use.  This work reports on a massive amount of data. It was analyzed in a sophisticated manner, is well-written, and provides valuable information. I must say that I really enjoyed reading it.

However, I do have some concerns, and I hope the following comments will help to improve the paper.

Title/Abstract

In my opinion, the title of the paper (and half of the discussion/conclusion) does not fit all the exciting results the authors have. Personality/temperament traits are part of a continuum that goes, in the case of fearfulness, from the least fearful individuals to the most fearful individuals. You did find that low fearful/proactive individuals (+ escape attempts, + vocalizations, - latency to the first step) spent more time in the range (L322-324 and L341-343), or I am completely wrong? I really do not understand why you state that fearfulness is not related to range use. For me, fearfulness is clearly linked to range use, based on your results.

21-22 – Proactive individuals spent more time in the range, I do not understand why the authors state the contrary here. See Koolhass et al. (1999) - Coping styles in animals: current status in behavior and stress-physiology - to confirm how your less fearful animals fit the proactive axis.

Introduction

46-47 – Please develop a little bit further why poultry may be particularly susceptible to fear of novel environments. Maybe adding some literature on junglefowl behavior.

51-52 – The paper cited is about fear in different species, and the examples are somehow mixed. Use some specific examples of fear in poultry or state which species you are referring to.

55-58 – I suggest moving the phrase “In experimental and commercial housing systems…” before “behavioral assessments”. This should facilitate reading.

88 – Although you mention the relationship between body weight and range use in a phrase (L67), I have the impression that the inclusion of body weight as one of the variables of interest appears out of nowhere. Maybe you should introduce this further. Or in a new phrase state that “Since body weight is also related to range use (citation), we also investigated….” or something similar.

Methods

113-117 – This division also separated the tested hens from the other 1875 hens (larger experiment)?

118 – Did you test all 769 chickens in one week?

132 – How do you target 300 hens per flock if in two flocks you have < 300 hens?

Figure 1 – There is no mention in the text to these 150 random individuals. You only mention that you targeted 300 per flock.

Behavioural tests – How many hens were tested per day? How many days of testing?

165 – Just to confirm, the arenas were also 100m from each other?

Table 3 – Just to remember and facilitate reading: remember that these results are from tested stayers and rangers combined.

Results

As I mentioned previously, your results during the NA test clearly show you have less fearful/proactive, and these individuals spent more time on the range (L322-324 and L341-343).

Discussion

365-366 – Can we really say that this way? The temperament may impact range and aviary use. I suggest the authors always say that there is an interaction/relationship, as they do in the next phrases.

369 – You did not mention this theory, and there is no citation. Please explain/rephrase. I feel like there is a dynamic relationship between curiosity and fearfulness. As the authors state, chickens seem to be less fearful over time, but also more curious.

The last three paragraphs of the discussion and parts of the conclusion follow the same pattern of what I mentioned for the title and abstract. Based on your results (L322-324 and L341-343), I consider that you have both proactive and reactive responses during the NA test, and these responses are linked to range use, which corroborates different papers on free-range laying hens coping styles (Larsen, Campbell etc…). So the authors did find evidence for a relationship fearfulness/coping styles and range use.

407-410 and 419-420 – Please develop more the social motivation part. Individuals are less numerous on the range during the beginning of range access, but over time this pattern changes, and more and more individuals go outside. The presence of many individuals on the range can motivate sociable individuals to visit the range. In a recent paper  it was showed that broiler rangers seemed to be more socially motivated.

Conclusion

438-439 – Your results (L322-324 and L341-343) do not support this statement.

Author Response

(The authors gave the same response as above.)

Reviewer 3 Report

I enjoyed reading this manuscript very much and consider the results relevant and important for the situation of free-ranging flocks. For the most part it is clearly written, and the methods and data are well presented in the tables and the supplements. I agree with the conclusion that this type of housing enables hens lower in hierarchy or more fearful birds to retreat to upper tiers in the aviary and this is good.

However, the experimental design remains unclear. Did the authors use 3 consecutive flocks, meaning they had 3 independent trials? Or were the hens just a composite of 3 different flocks and were tested simultaneously? In addition, I cannot understand why the hens were sorted and grouped into separate pens according to their early ranging behavior. If I understand correctly, the authors used just 2 pens (per trial or in total) and compared the birds between these 2 pens. Everybody working with laying hens knows that there are huge differences between pens without any difference in treatment. In that way, pen and ranging activity are confounded. This means that in the end the authors cannot attribute behavioral differences to the ranging behavior anymore, but it could also be a pen effect. Regrouping the hens at the start of laying also influences their social structure and the reason why anybody would want to do this is not clear to me.

Other details about the flocks are missing. Which hybrid was used? Was it the same hybrid in all 3 flocks? In which season did the study take place? What did the range look like? What kind of vocalizations did you record during the behavior tests? Alarm calls? This information might be found in the cited studies but it is very important for this publication.

I found the relationship between the early body weights and (ranging) behavior very interesting. This could indicate the importance of good rearing. Pullet management should achieve a high uniformity in body weights close to the target value by increasing access of feed, feed quality, etc. With this, the percentage of hens using the range might be increased. Too often, pullets at the end of rearing are too light and uniformity is poor.

Here are some minor points to consider:

Line 172: Which access door?

Figure 4B: The zones are not clear. Can you mark them with green lines?

Line 246: Of course. If you deduct a number from another number the difference will automatically correlate with one of the numbers. You can see that if you look at the formula.

Line 251f: I disagree that you can do that. Hens are such social animals influencing one another that I cannot see how you can take the individual hen as the experimental unit. Everybody working with chickens knows that the behavior of hens are very different among pens.

257f: This should be given when you first mention random forests.

Despite data reduction methods you still ended up with many variables which were correlated. Did you adjust the P-levels to the number of tests?

Line 333: This is probably due to the fact that the heavier hens started laying earlier.

Line 366: Why do you think that? It could also be the other way round. Their temperament might impact the use of the range and lower feeder.

Author Response

(The authors gave the same response as above.)
